# The Design and Fabrication of Large-Area Under-Screen Fingerprint Sensors with Optimized Aperture and Microlens Structures

**DOI:** 10.3390/s23218731

**Published:** 2023-10-26

**Authors:** Chih-Chieh Yeh, Teng-Wei Huang, You-Ren Lin, Guo-Dung Su

**Affiliations:** Graduate Institute of Photonics and Optoelectronics, National Taiwan University, Taipei 10617, Taiwan

**Keywords:** fingerprint on display, black matrix, absorbance, ray tracing, block layer

## Abstract

In this paper, we designed and fabricated an optical filter structure applied to the FoD (Fingerprint on Display) technology of the smartphone, which contains the microlens array, black matrix, and photodetector to recognize the fingerprint on a full touchscreen. First, we used optical ray tracing software, ZEMAX, to simulate a smartphone with FoD and a touching finger. We then further discussed how the aperture and microlens influence the fingerprint image in this design. Through numerical analysis and process constraint adjustment to optimize the structural design, we determined that a modulation transfer function (MTF) of 60.8% can be obtained when the thickness of the black matrix is 4 μm, allowing successful manufacturing using photolithography process technology. Finally, we used this filter element to take fingerprint images. After image processing, a clearly visible fingerprint pattern was successfully captured.

## 1. Introduction

In this era of digital information explosion, data security has always been an important issue. Biometrics has three advantages: uniqueness, universality, and permanence. It is widely used as a basis for the identification system. The biometric technology that is closest to our daily lives is facial recognition and fingerprint recognition applied to smartphones. Fingerprint recognition technology has a long history. In terms of smartphone applications, it has moved towards full-screen sensing in recent years. The primary purpose of this paper is to discuss the design simulation, manufacturing process, and measurement, which could be applied to the full-screen under-screen fingerprint recognition technology.

The fingerprint recognition method is commonly divided into three types: capacitive fingerprint [1], optical fingerprint [2], and ultrasonic fingerprint [3]. The capacitive type was used first, but this method is not suitable for full-screen and under-screen fingerprint recognition. The reason is that it cannot be placed under the screen due to its large space; the second reason is that large-area sensing requires many chips, leading to high costs. Ultrasonic induction also has the problem of high price. Therefore, we chose to use optical fingerprint recognition, keeping the image in view to penetrate the smartphone light modules, and putting an optical detector under the modules sensing the image.

The optical method benefits fingerprint detection but requires a high transmittance rate for all light modules. Because of the birth of the organic light-emitting diodes (OLED) panel in the last decade, we can choose a more transparent material for the substrate on the light modules of a smartphone that significantly increases the image’s brightness and the display’s transmittance rate, giving considerable support to the optical fingerprint recognition technology.

One of the immense challenges in large-area optical fingerprint recognition for OLED displays is the cost of the detection circuit. There are three common thin-film transistor technologies on the market, i.e., amorphous silicon (a:Si) [4], indium gallium zinc oxide (IGZO) [5], and low-temperature polysilicon (LTPS) [6]. As the first technology developed, a:Si TFT products have a lower reliability, aperture ratio, and PPI. In particular, the LTPS image sensor performs better than the IGZO image sensor in terms of PPI and has electron mobility and even advantages in manufacturing costs. However, compared to the on/off current ratio of the IGZO image sensor of up to 10^7^ [7], the LTPS image sensor has a current ratio of only about 10^5^ to 10^6^ [8,9,10], which leads to a worse photosensitive image in fingerprint image detection. If we want to solve this issue, careful optical design is necessary to compensate for the photosensitivity problem. 

K. Sasagawa. et al. proposed light guide arrays to solve this problem [11]. They successfully produced a silicon-based hole array with a high aspect ratio to limit the detection light incident angle and improve the image resolution. However, it is not easy to erode the sidewall of the hole to be completely vertical using etching technology. Compared to silicon-based light guide arrays, the black matrix made of Cr metal can be thinner, the side walls of the apertures are more vertical, and the light-blocking effect is better. Here, we used a black photoresist to make the aperture array. Although the result of the light blocking is slightly inferior to that of metal, it can reduce the reflected stray light [12]. The cost of making the black matrix using the lithography process can be lower. In addition to using the black matrix to block noise, microlenses have improved the optical signal under the screen [12,13,14,15,16]. When aligning with the microlens array and photodetector pixels, we can improve the optical signal in each pixel. There are many ways to manufacture microlenses. Thermal reflow is a relatively simple and inexpensive option that can be completed only by lithography [17,18]. To reduce the reflection of stray light at a large angle to the microlens, we added another aperture array layer between the aperture array and the microlens array to filter out the stray light. Based on the noise reduction of the double pinhole aperture array [19,20,21], we further combined the microlens array to increase the signal intensity received by the pixels of the sensor, which is very helpful for low light sensitivity. Moreover, the entire filter structure can be completed by lithography, which is quite simple, low-cost, and suitable for large-area manufacturing processes.

In this article, we simulate the fingerprint sensing module with a microlens array. The simulation model is divided into three modules: internal reflection, focusing, and filter. We focus on how to design the filter module. By adjusting the focal length of the microlens and the thickness of the black matrix, we analyze and calculate the MTF and ridge-valley (RV) values in different situations to optimize the structure. Experimentally, we also demonstrate how to use the lithography process to complete the microstructures and finally use it for fingerprint patterns and image analysis.

## 2. Simulation Methodology

### 2.1. Simulation Structure

The FoD simulation structures are divided into three parts: the smartphone screen, the microlens arrays, and the aperture filter layers. The whole structure is shown in Figure 1a.

The smartphone screen consists of several structural elements, including a sapphire substrate, a protective film layer covering the substrate, fingerprint simulation blocks attached above the protective film, and an OLED light source growing at the bottom of the sapphire substrate. Light incident at the bottom of the sapphire substrate passes through the protective film, impinges on the fingerprint simulation block, and reflects. The fingerprint simulation blocks comprise numerous 200 μm strips with 200 μm air gaps. These strips are set to be entirely absorbing to simulate the valley of our skin that touches the protective film, as the refractive index of our skin and the protective film are almost identical. By applying Fresnel’s equation, the phone module has virtually no reflection. We utilize absorbent strips to simulate no reflection and to prevent light scattering at a small intensity, thus saving simulation time. Additionally, an air gap is simulated to represent the valley of a fingerprint since the sag of a fingerprint typically contains air. It creates a scenario in which it is like air touching the protective film.

After the light impinges on the fingerprint simulation block, it reflects and carries the “fingerprint image signal” back into the smartphone, passing through the sapphire substrate and reaching the microlens component. The microlenses are designed with a back focal length near the photodetector, which enables the fingerprint signal to be emphasized into specific pixels. Mathematical analysis shows that this results in a more precise fingerprint image.

The final component, the aperture filter (pinholes), is designed to address the issue of the blurring of the fingerprint image. Although the fingerprint image can reflect on the surface of the protective film, it does not restrict the direction of reflection. As a result, the image can reflect at many different angles, all detected by the photodetector, resulting in a blurred image that cannot be recognized. The aperture filter is designed as two layers of a pinhole aperture, allowing only the correct angle (typical incident) to pass through the filter. By restricting the incident angle of the fingerprint, the detected image is recovered and is distinguishable.

This paper focuses on the lower part of the overall structure, where the microlenses and pinholes are located. We set the distance between the lenses at 36 μm to match the image pixel distance, as shown in Figure 1b. By designing the diameter and vertical distance of the microlens and pinhole, we can optimize fingerprint signals.

### 2.2. Optical Simulation and Analysis

To mimic the “fingerprint image” journey through a smartphone module before reaching the photodetector, we employ the ray-tracing function available in Zemax. Ray tracing involves calculating the path of a light beam originating from a light source. As the light beam travels, it may encounter reflection, refraction, or absorption by various material layers, causing alterations in its direction, intensity, and even polarization. When the light beam reaches the photodetector, it exhibits a particular intensity and phase distribution, which is documented and repeated with the light source emitting at different angles and positions tens of millions or billions of times. The average of all of these “radiative results from source to detectors” is then taken as the final light radiate outcome. This method is referred to as ray tracing. Figure 2a displays a 2D ray trace image (with blue lines representing light beams and simulating a single lens focusing light), while (b) illustrates the detector sensing image in the form of a 2D intensity diagram.

With ray tracing in Zemax, a 2D intensity distribution diagram is generated. However, when comparing two or more such images, it becomes challenging to determine their relative quality based only on visuals. To facilitate a more efficient comparison, objective “number data” are required. By exporting the 2D map intensity data of the diagram and converting them into an Excel database, we can then use Matlab to analyze the data using our mathematical formulas or calculations. This allows us to obtain objective indicators that can compare image results with more accuracy.

After performing ray tracing in Zemax, an original image similar to Figure 3a is obtained. The image appears to show uniformly distributed light dots throughout its entirety. However, upon closer inspection, one can observe that the beads vary in brightness and darkness while maintaining a uniform alignment. This information corresponds to the signal regarding the valleys and ridges. When testing multiple structures, each structure can produce an image similar to Figure 3a, making it challenging to compare all images due to their similarities. To overcome this issue, we process the image data into two indicators, RV and MTF, which facilitates easy comparison of results. The RV value represents brightness, that is, the amount of received signal; MTF is used to determine the signal quality of the RV (some will be noise). The higher the MTF value, the more precise and subtle the signal that can be detected. 

After importing the data into Matlab, the first step is to combine all horizontal data into one set, simplifying the “3D diagram” into a “2D diagram”, as shown in Figure 3b. This simplification facilitates the observation of the fingerprint signal. However, because of the structural design, there are gaps between each peak in the diagram. The next step is to remove these gaps and obtain a more simplified 2D diagram to address this issue.

To remove these gaps, we used a method to sum up the intensity of each peak and represent it as a new data point, as shown in Figure 4. Following these two steps of data processing, the fingerprint fringe becomes much more visible. Ultimately, we consider this fringe a natural optical fringe and calculate two indicators—RV and MTF—to serve as performance parameters for each image.

To distinguish between “bright fringe” and “dark fringe”, we require two representative values, one for each. We plot a “total average intensity” line on the diagram as a boundary between bright and dark fringes. We then identify the average intensity of both regions, representing the intensity values of the bright and dark fringes, respectively. We abbreviate these values: the bright intensity average corresponds to the fingerprint valley signal and is abbreviated as V, while the dark intensity average corresponds to the ridge signal and is abbreviated to R, as depicted in the schematic diagram shown in Figure 5.

Our RV and MTF are directly related to the intensity average “R” and “V”, but the calculation process is slightly different. RV is the difference between R and V, which follows that
RV = V − R(V is always larger than R)(1)

The value RV can express the intensity difference between the valley and the ridge signals. On the other hand, MTF follows the formula:(2)$$\mathrm{MTF}\;(\%)=\frac{V-R}{V+R}\times 100\%$$

Some may be confused that both parameters are similar at first glance, but they have different meanings in signal analysis. RV is used to calculate the absolute difference between bright and dark fringes; it relates to the concept that the exposure speed allows the detector to obtain sufficient image information. MTF is used to describe the quality of the fingerprint fringe, an idea commonly used for optical devices. If the parameter is high, the dark and bright fringes would be very clear and would have an exact boundary in the middle. On the contrary, if the MTF is low, the dark region would appear gray and not pure black; the bright area is also not obvious, and the boundary between them also seems blurred. Figure 6 shows the cross-comparison in the RV and MTF situations, which can help in understanding more about the RV and MTF parameters.

### 2.3. Simulation Results

Figure 1b shows how we use the LAA (lens aperture aperture) structure to strengthen the fingerprint signal. It is well known that lenses can gather signals, and apertures can block noise from different angles. We use these microstructures only to receive positive, near-vertical good signals.

But using a lens will encounter the aberration problem, and this structure’s spherical aberration is the most obvious. Through simulation, we found that by changing the lens’s radius curvature so that its focal length is slightly below the pinhole close to the detector, the signal collected at this time will be the best. The result is shown in Table 1.

After we set the focal length of the microlens as 31 μm, we need to determine the size of the micro aperture. We fixed aperture 1 to 3.5 μm and then adjusted the size of aperture 2 for the simulation; the results are shown in Table 2.

We found that the smaller aperture 2 is more helpful for MTF, and the larger aperture 2 is more helpful for RV, so we must find a balance between MTF and RV and proceed with the process. Based on the simulation conclusions and process limitations, we chose the 10 μm design because the MTF is maintained above 60% while the RV is close to 2. We believe that this value should be able to generate clear fingerprint images experimentally. 

## 3. Fabrication Processes and Experimental Results

To realize the structure of the multilayer aperture array and the microlens array, we adopted the photolithography technology of the semiconductor industry. Regardless of the microlens array (MLA) or the black matrix (BM), a photoresist is used as a material for lithography. First, regarding the microlens array, the type of photoresist we used is AZ4620 (from AZ Electronic Materials, Luxembourg). This is one of the photoresist types that is often used to make microlens arrays by thermal reflow. We adopted GMC1050 (Gersteltec, Pully, Switzerland) as the material to make the black matrix layer. It is a black resin hybrid of SU-8 photoresist and carbon grains and is characterized by high optical density (O.D.). The products made with it have a reasonable light-blocking effect. Finally, we filled the different microstructures with SU-8 2025 photoresist. It has the advantages of high transmittance in visible-to-near-infrared wavelengths, a high refractive index, good heat resistance, and mechanical resistance. Therefore, it can provide good support between different structural layers to prevent collapse. 

### 3.1. Process Steps

In order to have good light transmittance, we chose the glass substrate as the base to allow the structure to grow on it. Before the process started, we needed to clean the glass substrate. We soaked it in a solution of acetone, isopropanol, and deionized water in sequence while cleaning it with an ultrasonic oscillator for 10 min each. After drying the substrate with an air gun, we put it in a UV–ozone machine for 10 min. This work aims to increase the hydrophilicity of the surface of the glass substrate, making it easier for the photoresist to adhere to its surface. Once ready, we started our first photolithography process. The first structure to make is the black matrix. In the first step, we titrated GMC1050 on the glass substrate and set the rotation speed to 500 rpm for 10 s and then 3500 rpm for 30 s in two stages. The photoresist was spin-coated on the entire surface of the substrate. Then, we put it on a hot plate and heated it at 95 °C for 12 min. This action is called soft baking, which is used to accelerate the volatilization of the diluent in the photoresist. In addition to increasing the adhesion of the photoresist to the substrate, it also reduces the case where the photoresist sticks to the mask during subsequent exposure. After the substrate had cooled, it was exposed to UV light for 105 s. The pattern on the photomask was transferred to the photoresist. Then, we put the substrate on the hot plate and heated it again. The time and temperature settings were the same as those used for the soft bake. Through this post-exposure bake step, the exposed graphics can be made stronger. Finally, we soaked the substrate in the SU-8 developer for 1 min and stirred it slightly to speed up the development process. After drying it with an air gun, a black square array structure with a pinhole diameter of 3.5 μm and thickness of 4 μm was completed. The process of this part is shown in Figure 7.

Before making the second black matrix layer, we needed to lay SU-8 film with a thickness of 15 μm on the first layer of the black matrix. Its production method is the same as the above-mentioned lithography process steps, i.e., spin coating, soft baking, exposure, post-exposure baking, and development. However, before coating the SU-8 photoresist on the first layer structure, we must put the sample into the UV–ozone machine for 10 min. Through this work, the hydrophilicity of the surface can be increased, and the photoresist adheres more easily to the previous structure during coating. It should be noted that this process will be completed before the subsequent stacking of different structures. There are two differences in making an SU-8 film and a black matrix. The first one is that the selected photoresist is different, so the process parameters are different. The second is that when making SU-8 film, there is no need to use a patterned mask for exposure. The process method of the second black matrix is almost the same as that of the first one; their thickness specifications are the same, but since the aperture diameter of the second black matrix is 10 μm, the process parameters used are slightly different. An essential part to note here is that when exposing, it is necessary to pay attention to completely align the aperture center of the second black matrix with the aperture center of the first one. If the center deviates too much, the focus effect of the produced microstructure will deteriorate, and the fingerprint image will also become blurred. After the two layers of the black matrix were completed, it was also covered with a layer of SU-8 film to facilitate the subsequent production of microlens arrays. The method and parameters of the SU-8 film were exactly the same as those mentioned above. The overall fabrication parameters are completely organized into the following Table 3.

The last structure to be fabricated was the microlens array, which we chose to implement in this paper using thermal reflow. Its principles and steps were quite simple. First, the AZ4620 photoresist was spin-coated on the surface. We set the two-stage rotation speed as 500 rpm for 10 s and then 1300 rpm for 30 s. Next, we put it on a hot plate and heated it up for 10 min at 100 ° C. After the substrate had cooled, exposure was performed for 50 s. Similarly, it was also necessary to pay attention to whether the center of the circular hole of the pattern on the mask was aligned with the center of the black matrix aperture below. Then, we placed the sample in a container of developer solution for 3 min and shook it slightly. The developer solution was made by mixing AZ400K and deionized water at a ratio of 1:3. After completing the above steps, the columnar array structure was obtained on the previous multi-layer structure. The so-called thermal reflow process involves heating the cylindrical islands array structure, and we set the temperature at 150 °C for 50 s. Because the heating temperature exceeded the glass transition temperature of the AZ4620 photoresist, the cylindrical islands began to melt, and due to the balance of the surface energy, it gradually became a shape similar to a spherical cap, that is, a microlens. Through theoretical calculations based on our previous work [22], the radius of theoretical value of the curvature of the lens was about 18.6 μm. The entire thermal reflow process is shown in Figure 8.

### 3.2. Fingerprint Images

After completing the entire micro-optical element, we needed to encapsulate it, in addition to preventing the microlens from becoming dirty or damaged, but also as an interface for fingerprint recognition, allowing us to place our fingers on the glass. Therefore, we needed to fill in spacers between a piece of glass and the micro-optical structure for support. We set up an experiment to capture the imaging situation of real fingerprints passing through this structure. Because we could not obtain photoelectric sensors and OLED panels, we used CCD cameras instead of photoelectric sensors and general panel LED lights as light sources. We placed an objective lens and tube lens on the back of the structure to extend the focal length of the CCD camera, as shown in Figure 9. Since the LED light source cannot be placed directly behind the micro-optical structure, it will block the reflected light of the fingerprint and make the CCD camera unable to capture the image. Therefore, the light source had to enter from the side to shoot the image, but this also led to uneven brightness of the image.

Due to the problem of the incident angle of the light source, the brightness distribution of the image was very uneven and the fingerprint pattern was not very clear or identifiable. Therefore, we further used the program to process the fingerprint image in grayscale and removed high-frequency noise through the Fourier transform. Then, we binarized the image. Finally, we used the computer to remove the fingerprint lines for thinning processing to confirm that the image we captured was indeed the fingerprint pattern, as shown in Figure 10.

## 4. Conclusions

In this paper, we improved the image clarity of optical under-screen fingerprint recognition by adding a micro-optical structure between the sensor and the panel. In addition to the glass substrate, the micro-structure mainly includes the microlens array and double-layer aperture array. The purpose of the microlens array is to focus the fingerprint signal on a specific photosensor pixel to enhance the strength of the signal. The aperture arrays are used as a black matrix, allowing light to pass through tiny holes, reducing unwanted noise, and making the image clear enough for biometric applications. 

To realize this design, we used exposure lithography technology from the semiconductor industry to stack the microstructure layer by layer, and the critical microlens array layer was fabricated using the technique of thermal reflow. Through continuous testing and experimentation, we finally found materials and process parameters suitable for the design specifications and produced the microstructure. We also successfully set up optical experiments to experimentally capture fingerprint images and used computer programs to convert the obtained images into binary images. It is confirmed that the fingerprint imaging can be identified. The proposed microstructures in this paper have the potential to be used as full-screen under-display fingerprint sensors. 

## Figures and Tables

**Figure 1 sensors-23-08731-f001:**
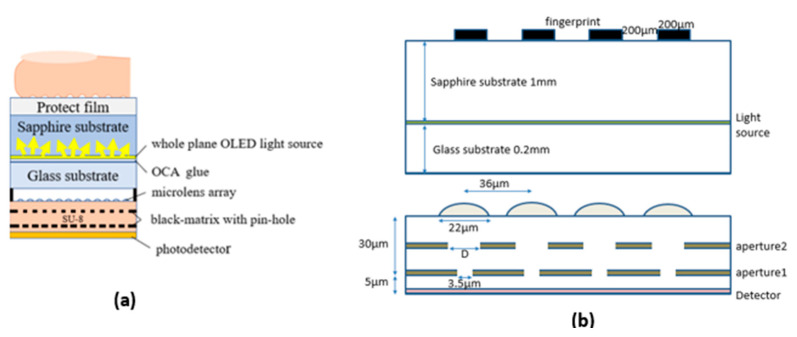
The simulation structure. (**a**) The structure of mobile phone fingerprint recognition. (**b**) The detailed simulation structures of FoD.

**Figure 2 sensors-23-08731-f002:**
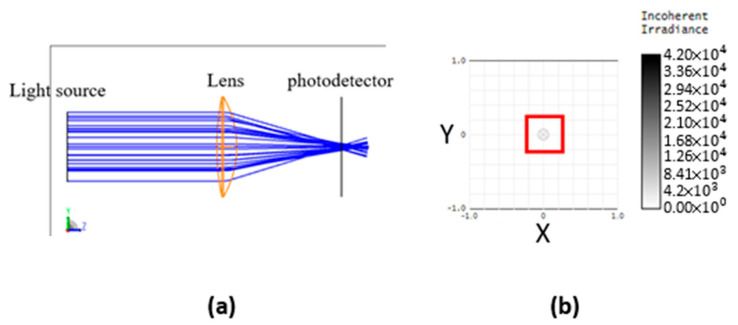
Ray tracing. (**a**) 2D structure viewing. (**b**) Detector image sensing a small focusing spot.

**Figure 3 sensors-23-08731-f003:**
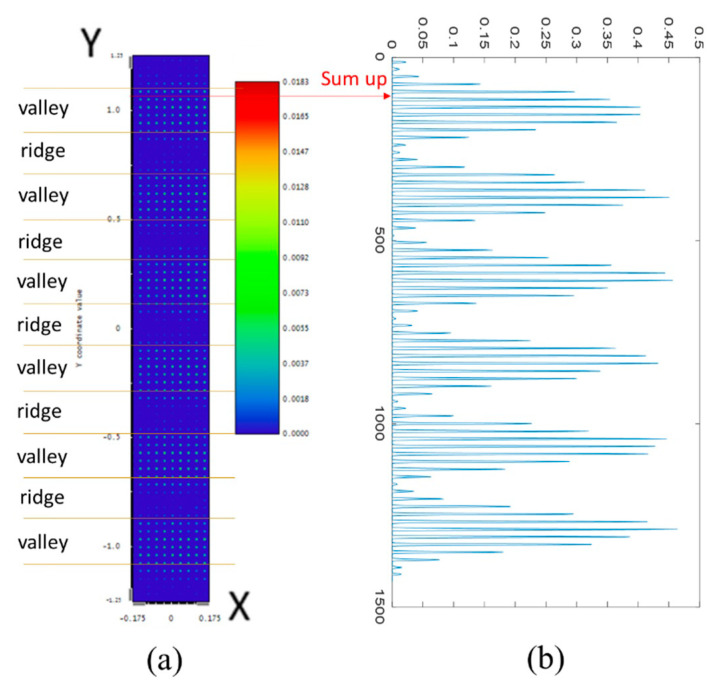
Processing data. (**a**) Ray tracing image in Zemax, (**b**) first process, i.e., “sum up all intensity data in the same horizontal line”, turned into a 2D diagram; the vertical axis represents the length (μm), and the horizontal axis represents the intensity (A.U.).

**Figure 4 sensors-23-08731-f004:**
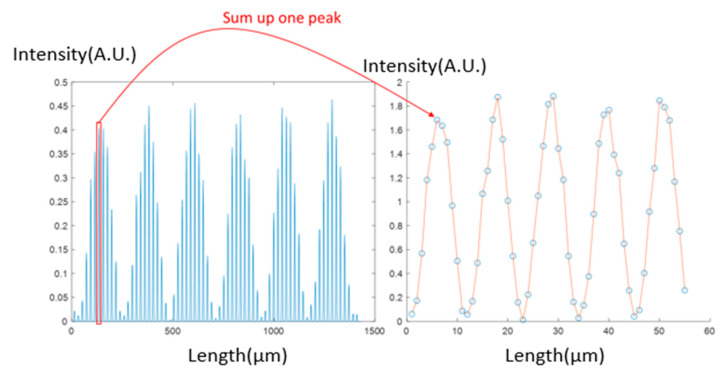
Data processing of summing up the peaks.

**Figure 5 sensors-23-08731-f005:**
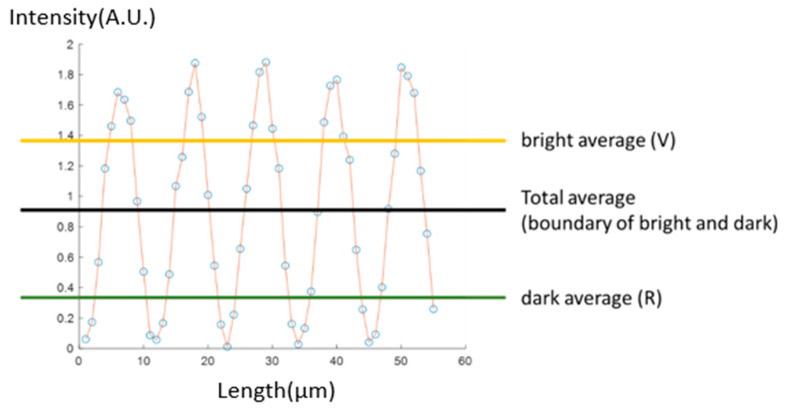
Schematic diagram showing R and V.

**Figure 6 sensors-23-08731-f006:**
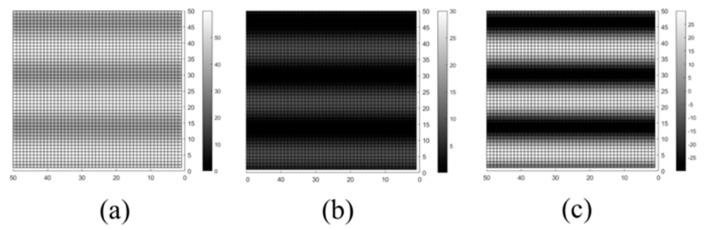
RV and MTF cross-comparison. (**a**) High RV but low MTF. (**b**) High MTF but low RV. (**c**) RV and MTF are high.

**Figure 7 sensors-23-08731-f007:**
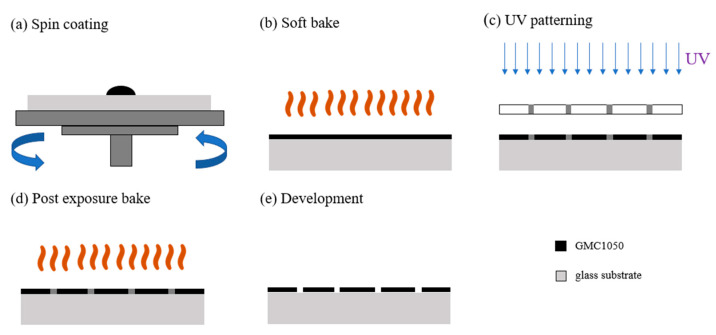
The process flow chart of the first layer of the black matrix.

**Figure 8 sensors-23-08731-f008:**
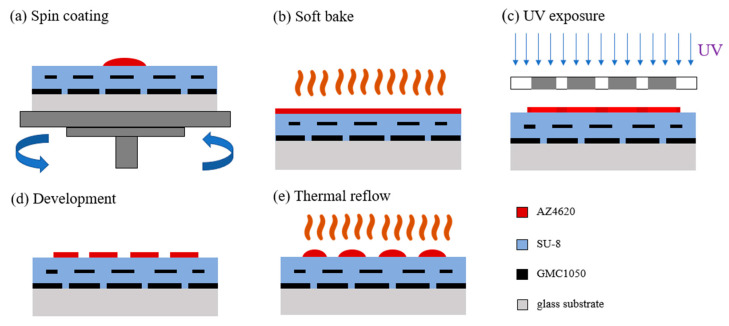
Flow chart of microlens array manufacturing using the thermal reflow method.

**Figure 9 sensors-23-08731-f009:**
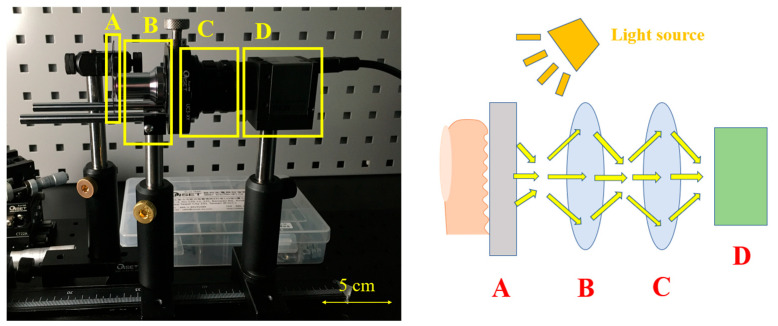
Fingerprint photo experiment: (A) micro-optical structure, (B) objective lens, (C) tube lens, and (D) CCD camera.

**Figure 10 sensors-23-08731-f010:**
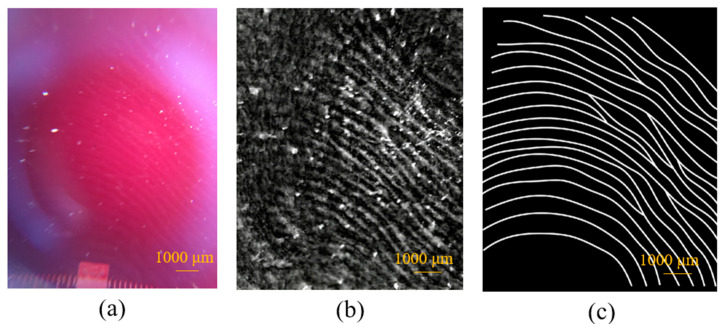
(**a**) The image of the fingerprint captured by the smartphone, (**b**) the image processed in grayscale, and (**c**) the thinning the fingerprint texture by a computer program.

**Table 1 sensors-23-08731-t001:** Simulation and results of different focal lengths.

Focal Length (μm)	23	25	27	29	31	33
RV	1.2428	1.3629	1.5213	1.6428	1.6892	1.6271
MTF	52.71%	54.63%	57.42%	62.57%	67.83%	63.41%

**Table 2 sensors-23-08731-t002:** Simulation and results of different aperture 2 values.

Aperture 2 (μm)	7	8	9	10	11	12
RV	1.6972	1.5916	1.7381	1.9214	2.0743	2.2154
MTF	68.71%	65.47%	64.76%	61.48%	58.27%	56.42%

**Table 3 sensors-23-08731-t003:** Process parameters used in the first three layers.

	First BM	SU-8 Film	Second BM
Spin coating	500 rpm @ 10 s and 3500 rpm @ 30 s	500 rpm @ 10 s and 1600 rpm @ 40 s	500 rpm @ 10 s and 3500 rpm for 30 s
Soft bake	95 °C @ 12 min	95 °C @ 35 min	95 °C @ 15 min
Exposure time	105 s	90 s	170 s
Post exposure bake	95 °C @ 12 min	95 °C @ 30 min	95 °C @ 15 min
Development time	1 min	2 min	80 s

## Data Availability

Not applicable.

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
