# Peer review of "The Design and Fabrication of Large-Area Under-Screen Fingerprint Sensors with Optimized Aperture and Microlens Structures"

_sensors, 2023, doi:10.3390/s23218731_

Round 1

Reviewer 1 Report

Manuscript: Design and fabrication of large-area under-screen fingerprint sensors with optimized aperture and microlens structures.

The authors have written a very nice manuscript regarding an approach and part of the solution for a fingerprint-on/in-display application for smartphones. This is also a hot topic for upcoming smartphone models and I would recommend publishing after some changes and possible additions to improve the quality of the manuscript. Below my remarks and/or additions:

1. Line 43-54. This is a small reference to image sensors. Unfortunately the statements are not correct. An image sensor consists of a frontplane (the photodiode pixels) and backplane (the TFT driving circuitry).  The TFT in general will be oxide-based (eg IGZO), LTPS or a-Si. Hence, the first reference to organic image sensor is compared LTPS or IGZO does not make any sense, they refer to a different part of the technology. For example reference 4 has an IGZO backplane too but with an organic frontplane. for example reference 6 explains:

"The structure of the transistor used in this sensor is the same as those used in a display TFT backplane, and is formed on a glass substrate. Difference from a display panel in this prototype is that this device has a photodiode on top instead of a light-emitting device. The processing circuitry in the pixel of this work is independent from the photodiode, and thus there is no limitation on the type of photodiode technology to be used. For example, the photodiode may be a selenium-based30) or an organic-material-based device. If the photodiode can be fabricated in the back-end-of-line processes, a large aperture ratio for each pixel becomes possible."

This holds for all the references in this section and means it needs to be re-written to be correct. Please keep in mind that the on/off ratio is determined by the frontplane diode technology, but this is different than dark leakage which is determined by the frontplane or the backplane. The on/off ratios in my version are given by 105 or 106, instead of 10^5 or 10^6. I assume this will be correct in the manuscript original. 

2. The model for the fingerprint is now 200 um lines with 200 um pitch. A fingerprint is much more complicated than that, but it is a good starting point.

3. The pitch between lenses is 36 um. In the modelling the focal length of the lens is changed. It should say something that the pitch is matching an image sensor pitch (not sure why 36 um is fixed since no image sensor is used). It should also provide numbers for lens diameter and sag or radius of curvature.

4. From the modeling it would be interesting to see comparison graphs: for example, with and without the aperture 1 layer, is it needed? What if we vary pinhole dimensions etc. 

5. Currently the layer stack is completed with lenses made with a reflow of AZ4620. However, it does not have any characterization of the resulting lens, i.e. does it have the right curvature/sag, microscopy image, ...

6. Instead of a picture of the setup in figure 10, please also add a schematic. For example, the CCD camera will have a certain focal depth which is very different compared to an image sensor. How does this affect the results?

7. Finally, an MTF is obtained from the modelling for a fixed line pair width value. This is possible to do for a fair comparison. However, an MTF is useful since it describes the contrast values as a function of optical target resolution. This way also something can be said about all the smaller features present in a fingerprint.  

Reviewer 2 Report

The paper “Design and fabrication of large-area under-screen fingerprint sensors with optimized aperture and microlens structuresdesigned and fabricated an optical filter structure applied to the FoD (Fingerprint on Display) technology of the smartphone. There are some doubts:

1. In lines 223-226, Why choose a 10μm aperture? and what is the rationale behind this choice? Why wasn't 9μm or 11μm selected? In seeking a balance between MTF and RV, based on the formula mentioned earlier, RV directly influences MTF. Why not directly find the optimal RV?"

2. The data shown in Figure 3(b), Figure 4, and Figure 5 do not indicate the specific name of the horizontal and vertical coordinates, which is not particularly clear.

3. The three diagrams in Figure 10. The lower right corner of a shows 690 microns, and b and c both show 1000 microns. What do these numbers represent? Can explain clearly

 Minor editing of English language required

Round 2

Reviewer 1 Report

The authors have done a very good job with implementing the proposed recommendations, I also do think the paper is of better quality now and more complete. I would suggest to publish as is.